# Somatic Mutations in Exon 7 of the *TP53* Gene in Index Colorectal Lesions Are Associated with the Early Occurrence of Metachronous Adenoma

**DOI:** 10.3390/cancers14122823

**Published:** 2022-06-07

**Authors:** Tereza Hálková, Renata Ptáčková, Anastasiya Semyakina, Štěpán Suchánek, Eva Traboulsi, Ondřej Ngo, Kateřina Hejcmanová, Ondřej Májek, Jan Bureš, Miroslav Zavoral, Marek Minárik, Lucie Benešová

**Affiliations:** 1Centre for Applied Genomics of Solid Tumors (CEGES), Genomac Research Institute, Drnovská 1112/60, 161 00 Prague, Czech Republic; thalkova@genomac.cz (T.H.); rptackova@genomac.cz (R.P.); asemyakina@genomac.cz (A.S.); mminarik@elphogene.cz (M.M.); lbenesova@genomac.cz (L.B.); 2Department of Medicine, 1st Faculty of Medicine, Charles University and Military University Hospital Prague, U Vojenské Nemocnice 1200, 169 02 Prague, Czech Republic; miroslav.zavoral@uvn.cz; 3Department of Gastrointestinal Oncology, Military University Hospital Prague, U Vojenské Nemocnice 1200, 169 02 Prague, Czech Republic; bures.jan@uvn.cz; 4Department of Pathology, Military University Hospital Prague, U Vojenské Nemocnice 1200, 169 02 Prague, Czech Republic; eva.traboulsi@uvn.cz; 5Faculty of Medicine, Institute of Biostatistics and Analyses, Masaryk University, Kamenice 126/3, 625 00 Brno, Czech Republic; ngo@iba.muni.cz (O.N.); hejcmanova@iba.muni.cz (K.H.); majek@iba.muni.cz (O.M.); 6Elphogene, Drnovská 1112/60, 161 00 Prague, Czech Republic; 7Department of Analytical Chemistry, Faculty of Science, Charles University, Hlavova 2030/8, 128 00 Prague, Czech Republic

**Keywords:** colorectal cancer, colorectal adenomas, colonoscopy, index lesion, synchronous lesion, metachronous lesion, tumor heterogeneity, *TP53*

## Abstract

**Simple Summary:**

Identifying patients with an increased risk of early recurrence of colorectal lesions is still a problem. In our study, we focused on improving this identification by determining the mutation profile of index lesions. We found a statistically significant association between the mutation in exon 7 of the *TP53* gene in the index lesion and the risk of early metachronous adenoma.

**Abstract:**

(1) Background: this prospective study was focused on detailed analysis of the mutation heterogeneity in colorectal lesions removed during baseline (index) colonoscopy to identify patients at high risk of early occurrence of metachronous adenomas. (2) Methods: a total of 120 patients after endoscopic therapy of advanced colorectal neoplasia size ≥10 mm (index lesion) with subsequent surveillance colonoscopy after 10–18 months were included. In total, 143 index lesions and 84 synchronous lesions in paraffin blocks were divided into up to 30 samples. In each of them, the detection of somatic mutations in 11 hot spot gene loci was performed. Statistical analysis to correlate the mutation profiles and the degree of heterogeneity of the lesions with the risk of metachronous adenoma occurrence was undertaken. (3) Results: mutation in exon 7 of the *TP53* gene found in the index lesion significantly correlated with the early occurrence of metachronous adenoma (log-rank test *p* = 0.003, hazard ratio 2.73, 95% confidence interval 1.14–6.56). We did not find an association between the risk of metachronous adenomas and other markers monitored. (4) Conclusions: the findings of this study could lead to an adjustment of existing recommendations for surveillance colonoscopies in a specific group of patients with mutations in exon 7 of the *TP53* gene in an index lesion, where a shortening of surveillance interval may be warranted.

## 1. Introduction

It is well known that the incidence and mortality of colorectal cancer (CRC) is rising worldwide. Predictions of incidence and mortality show a continuing increase between years 2020 and 2040 for colon cancer by 55% and 70%, respectively, and for rectal cancer by 50% and 66%, respectively [1]. An important tool for reducing the incidence of colorectal cancer is screening followed by removal of precancerous lesions during index or surveillance colonoscopies [2]. Although there are guidelines for surveillance intervals based on clinical and histopathological characteristics of the lesion that classify patients as high- or low-risk, they are not optimal and are difficult to follow in practice.

In general, short surveillance intervals lead to the exceedance of the capacity of the colonoscopic units, which is followed by a waiting time prolongation, a decrease in patients’ compliance, and an increase in the healthcare cost. Therefore, the general trend is to extend surveillance colonoscopy intervals. On the contrary, longer intervals might increase the risk of developing more severe colorectal neoplasia. According to current European Society for Gastrointestinal Endoscopy (ESGE) guidelines updated in 2020 [3], surveillance colonoscopy intervals are 10 years for patients with a low risk and 3 years for patients with a high risk of colorectal neoplasia development. The low-risk patients have been defined as those with 1–4 adenomas, size <10 mm with low-grade dysplasia (LGD), or any serrated lesion size <10 mm with no dysplasia. High risk has been defined as the presence of at least 1 adenoma sized ≥ 10 mm and/or with high-grade dysplasia (HGD), or ≥5 adenomas, or any serrated lesion sized ≥ 10 mm or with dysplasia. However, in addition to the classical morphological features used to estimate the risk of metachronous lesion recurrence, the presence of somatic mutations in the index lesions could be equally important. In recent years, research on colorectal cancer and its premalignant lesions has increasingly progressed towards determining the presence, localization, and spread of genetic changes, mainly somatic mutations, within the tumor [4,5,6,7,8]. Mutations characterize the individual intratumor clones, especially in terms of their origin, development in space and time, and interrelationships with the environment. Most importantly, from a clinical point of view, mutations also determine the various biological properties of tumors. They influence the response to various types of treatment and drug resistance [9,10], course of the disease, recurrence, and ability to form metastases [11,12]. While the occurrence of mutations in early and advanced CRC is well mapped, heterogeneity of mutation profiles in index lesions is much less studied.

The major aim of our current study was to map in detail the mutational heterogeneity of colorectal lesions detected by index colonoscopy to help better identify patients at higher risk of early recurrence of metachronous lesions.

## 2. Materials and Methods

### 2.1. Patients

This study was focused on 120 patients with at least one colorectal lesion ≥10 mm removed during index colonoscopy and undergoing surveillance colonoscopy after 10–18 months. All colorectal adenomas were sporadic, not familial ones in all cases. The cohort of 120 patients included 74 men, and 46 females, aged from 34 to 75 years, mean of 61, median of 64 years.

Detailed clinical, endoscopic, histopathological, and laboratory data were collected from all patients, including gender, age at diagnosis, number, size, location, morphological type, and degree of lesion dysplasia according to valid WHO criteria (World Health Organization Classification of Tumors, 2010) and method and radicality of resection.

The study complied with the ethical standards of the World Medical Association’s Declaration of Helsinki. All patients signed informed consent to provide their clinical data as well as biological samples for molecular analyses. The study was approved by the Ethics Committee of the Military University Hospital, Prague, Czech Republic (Protocol number 108/9-59/2016) and the study protocol was registered at ClinicalTrials.gov with ID: NCT03434925. For all data obtained, all personal identification information was removed in compliance with the European Union Regulation 2016/679 (General Data Protection Regulation).

### 2.2. Index and Surveillance Colonoscopy

Index and surveillance colonoscopies were performed at the Department of Gastrointestinal Endoscopy, Military University Hospital. All lesions ≥ 10 mm (regardless of other histopathological characteristics) found during an index colonoscopy were considered as index lesions and removed in one piece using en-bloc resection (endoscopic polypectomy, endoscopic mucosal resection, or endoscopic submucosal dissection). Other lesions found during index colonoscopy were considered as synchronous lesions. Surveillance colonoscopy was performed 10–18 months after index colonoscopy. All adenomas found at surveillance colonoscopy were considered as metachronous adenomas.

### 2.3. Samples

Histopathological analysis was performed at the Department of Pathology, Military University Hospital. Each index lesion was vertically cut into 2–7 parts and after formalin fixation and paraffin-embedding photo documentation was performed. Other lesions (synchronous and metachronous) were treated with standard procedures. All formalin-fixed paraffin-embedded (FFPE) blocks were provided to the molecular genetics laboratory.

### 2.4. Molecular-Genetic Analysis

All index lesions (each cut into 2–7 paraffin blocks) and synchronous lesions (each cut into 1–5 paraffin blocks) were provided for molecular heterogeneity analysis. In the case of index lesions, each paraffin block was further divided into smaller parts so that the total volume of the analyzed tissue sample corresponded to a maximum of 5 mm^3^ in order to capture intratumoral heterogeneity in as much detail as possible. Synchronous lesions were subjected to less thorough analysis—each paraffin block was divided into samples with a volume of about 20 mm^3^. DNA was extracted from each sample using a Gen Elute FFPE DNA Purification kit (Sigma, St. Louis, Missouri, USA) according to the manufacturer’s instructions.

DNA was subjected to mutation analysis using a panel of 11 target hot spot mutation regions according to the Catalogue of Somatic Mutation in Cancer, COSMIC [13]. Specifically, it was *APC* exon 15—the mutation cluster region (MCR), *TP53* exons 5–8, *KRAS* exon 2, *PIK3CA* exon 9, and *BRAF* exon 15. Details of the mutation assay were described previously [14]. The analysis was performed using the PCR-based method with heteroduplex formation followed by their separation by denaturing capillary electrophoresis (DCE) at an optimal temperature in ABI Prism 3100 (Applied Biosystems, Waltham, Massachusetts, USA) as described previously [15]. This technique allows highly sensitive detection of even a small number of mutated alleles in an excess of non-mutated alleles. The hetero-duplex analysis results were visualized using Gene Marker v2.4.2. From the resulting electropherogram, the percentage of individual mutated alleles was determined from the peak heights as shown earlier [15].

### 2.5. Detection of TP53 Mutations in Metachronous Adenomas

Similarly to synchronous lesions, metachronous adenomas were divided into samples with a volume of about 20 mm^3^ and mutation analysis of exons 5, 6, 7, and 8 of the *TP53* gene was performed in each sample using the same technique as described above.

### 2.6. Determination of the Degree of Heterogeneity

Based on the number and percentage of individual mutated alleles across all samples of each index lesion, the degree of heterogeneity of the index lesions was determined according to Table 1. The percentage difference in the mutated fractions was equal to the difference between the parts of the lesion with the lowest and highest percentage of the mutation. For example, if the *KRAS* mutation was present in three samples of a given index lesion at 4%, 13.5%, and 18%, the difference in the mutated fractions was 14%. If 2 or more mutant clones were present in the lesion (2 or more different mutations were found), the clone with the highest percentage difference in mutated fractions was used to determine the degree of heterogeneity. Three degrees of mutation fraction differences were defined:

Highest—lowest percentage < 35%     low difference

Highest—lowest percentage ≥ 35% < 70%  medium difference

Highest—lowest percentage ≥ 70%     high difference

### 2.7. Statistical Analysis

A statistical analysis to determine the correlation between the occurrence of metachronous adenomas and the mutation profile or degree of heterogeneity of index lesion(s) was performed on a group of patients with all available input variables. Statistical analysis of the relationship between the mutation profile of lesions removed during index colonoscopy and the incidence of metachronous adenomas was further extended to include synchronous lesions. Survival curves showing the occurrence of metachronous adenomas were estimated by the Kaplan–Meier method. Comparison of curves was conducted by the log-rank test and by the Cox proportional hazard model adjusted for age and sex to estimate the hazard ratio. *p* values ˂ 0.05 were considered statistically significant. The data were processed in the Stata/IC 14 software (StataCorp, College Station, TX, USA).

## 3. Results

### 3.1. Surveillance Colonoscopy Results

Of the 120 patients, 52 (43%) had no lesion on a surveillance colonoscopy. Of the remaining 68 patients, 15 (22%) had hyperplastic polyps and 53 (78%) adenomas (8 advanced and 45 non-advanced). The majority of adenomas (39; 74%) were diminutive adenomas, only 14 (26%) adenomas were larger than 5 mm. No colorectal cancer was detected.

### 3.2. Mutation Analyses of Index and Synchronous Lesions

In a group of 120 patients 11 patients had more than 1 index lesion—4 patients had 2 lesions, 4 patients had 3 lesions, 1 patient had 4 lesions, and 2 patients had 5 lesions. Thus, a total of 143 index lesions (1325 samples) were analyzed. A complete mutation profile from all samples was obtained in 104 index lesions originating from 83 patients. The number of mutation clones in each lesion varied (see Table 2). Detailed results of mutation analysis of index lesions in all 83 patients and results of surveillance colonoscopy are shown in Appendix A.

In 66 patients with 76 index lesions, the percentage of mutated fractions across all samples was successfully determined and allowed to determine the degree of heterogeneity within a given lesion. Thus, the lesions were classified into 5 degrees of heterogeneity with the following representation: grade I (*N* = 29; 38%), grade II (*N* = 14; 18%), grade III (*N* = 5; 7%), grade IV (*N* = 15; 20%), and grade V (*N* = 13; 17%). In the remaining 28 lesions of 17 patients, it was not possible to quantify the percentage of the mutated fractions in all samples and thus reliably determine the degree of heterogeneity.

In this group of 66 patients, 39 patients had synchronous lesions (a total of 84, divided into 117 samples). In 4 synchronous lesions of 4 patients, mutational analysis could not be performed due to DNA degradation. For more details, see the flow-diagram in Figure 1. The numbers of mutations found in the remaining 80 lesions are summarized in Table 2. No more than 2 mutations were found in any lesion.

### 3.3. Statistical Analyses

In the same group of 66 patients, there were 36 patients with metachronous adenomas. A statistical analysis of the association between the occurrence of metachronous adenomas and the mutation profile of the index lesion as well as the degree of heterogeneity was performed. The relationship between the presence of a mutation in exon 7 of the *TP53* gene in the index lesion and the occurrence of metachronous adenomas proved to be statistically significant (*p* = 0.003, log-rank test; hazard ratio 2.73, 95% confidence interval 1.14–6.56), see Figure 2.

Mutations in exon 7 of the *TP53* gene were found in 8 index lesions of 8 patients, all of whom had metachronous adenomas. Interestingly, none of the metachronous adenomas in this group of patients had any *TP53* mutation. For more details see Table 3. Other tested variables including tumor suppressors, oncogenes, individual genes, gene loci (including other *TP53* exons), as well as the degree of heterogeneity of index lesions, were not shown to be statistically significant predictors of the risk of metachronous adenomas.

Statistical analysis of the correlation of the mutation profile and the incidence of metachronous adenomas was further performed on a set of all lesions found in 66 patients during index colonoscopy (index lesions + synchronous lesions). This analysis confirmed all the results of the original analysis. Since no mutation in exon 7 of the *TP53* gene was found in any synchronous lesion of any of the 66 patients, the statistically significant relationship between the mutation in exon 7 of the *TP53* gene present in the index lesion and the occurrence of metachronous adenoma was not modified.

## 4. Discussion

The number of mutations in colorectal precancerous lesions has been shown to be the same, if not higher, than in carcinomas [16,17,18]. In addition, the frequency and distribution of mutations in precancerous lesions play an important role in their development into malignancy [19]. Therefore, there is the question of whether a detailed mutation profile of index lesions could help identify patients at high risk of lesion recurrence.

In this study, a detailed analysis of mutation heterogeneity by systematic sampling and genetic profiling of many areas of the same lesion was performed. The methodological approach used was unique in three aspects. First, all lesions ≥ 10 mm found during index colonoscopy were examined. Secondly, the index lesions were removed only by en-bloc resection; therefore, the specimen remained in one piece. This procedure is associated with a low to zero risk of local recurrence in the scar. Third, patients were followed with surveillance colonoscopy at a shorter interval than usual—from 10 to 18 months with focus on the presence of early metachronous adenomas.

The number of mutational clones found in the index lesions varied, but most often one (in 42% of the lesions), none (26%), or two (17%) mutation clones were found within one index lesion. The presence of mutation clones was as we expected lower in synchronous lesions—no mutation was detected in 55% of them, the remaining lesions had mostly 1 mutation (37.5%) and a maximum of 2 mutations (7.5%). This was due to the more advanced stage of index lesions compared to synchronous lesions.

Mutations were found in different parts of the lesion, in different numbers of parts of the lesion and in different percentages. For easier orientation in the mutational heterogeneity, a degree of heterogeneity was determined, which takes into account the “total mutation load”—i.e., the number of mutational clones and their percentage in different parts of the lesion. The higher the number of mutations and/or the greater the difference in the percentage of mutated alleles, the higher the degree of heterogeneity. Surprisingly, however, no statistical relationship was found between the degree of heterogeneity of an index lesion and the early onset of metachronous adenoma. This seems to depend on the presence of a particular mutation rather than on the “mutation load”.

As the results show, the mutation in exon 7 of the *TP53* gene is statistically related to the risk of metachronous adenoma, so patients harboring this mutation in index lesions likely have a higher risk of early metachronous adenoma occurrence. Adenoma was detected in all 8 of them during a surveillance colonoscopy. In this group of patients, we did not observe any noticeable differences from patients with other/no mutations in terms of a higher number/size/degree of dysplasia of the lesions in our study. In fact, all 8 patients with the *TP53* exon 7 mutation are very diverse in this respect, and apart from the occurrence of metachronous adenoma in a short time interval, we did not observe much in common. Thus, the mutation in exon 7 of the *TP53* gene itself seems to predispose individuals to the occurrence of metachronous adenoma but by what mechanism or manner remains unclear. For example, a mutation in exon 7 of the *TP53* gene could be caused by some kind of general molecular defect that affects the growth of metachronous lesions. Or, perhaps, there may be a link between a mutation in exon 7 of the *TP53* gene and the patient’s lifestyle, where the finding of this mutation as well as the occurrence of metachronous adenomas may be a manifestation of the same unhealthy behavior. Interestingly, mutations in other of the tested exons of the *TP53* gene, as well as mutations in the *TP53* gene in general, were not statistically significant predictors of the occurrence of metachronous adenoma.

We are aware of possible limits of our study. The question arises as to whether the adenomas found in surveillance colonoscopy in patients with the *TP53* exon 7 mutation in the index lesion were fast-growing metachronous lesions or interval lesions missed by index colonoscopy. In our cohort of 66 patients, such subsequent adenomas in one-year surveillance colonoscopy were found in 36 of them (54.5%), which is in agreement with another recently published study [20]. These adenomas were found in a different location than the index lesion, so they were not considered local residual neoplasia (LRN) in the post-resection scar. Moreover, none of the metachronous adenomas had the *TP53* exon 7 mutation detected in the original index lesion. However, the question remains whether this mutation would not appear later in more advanced adenomas. Fully reliable, indisputable distinction between early metachronous lesions and possible interval sporadic colorectal neoplasia at a surveillance colonoscopy is difficult and demanding. To solve this issue, shorter interval between index and surveillance colonoscopy (1 year instead of 3 years recommended) were chosen. Diminutive polyps (<5 mm in size) are most probably early metachronous lesions. Advanced polyps (≥10 mm in size) might be interval lesions missed by index colonoscopy. We did not find any interval colorectal cancer. Another possible limit of our study is that we did not consider histopathology and left- and right-sided neoplasia, as the subgroups would be asymmetric in numbers of subjects for appropriate evaluation. It should further be emphasized that our analyses were performed with rather small sample sizes. Of the entire set of 104 index lesions and 80 synchronous lesions with complete mutation profiles, mutations in exon 7 of the *TP53* gene were found in only 8 index lesions of 8 patients. It is therefore necessary to take into account the possibility that a larger sample size could give different results.

The tumor suppressor gene *TP53* is the most frequently mutated gene across 12 major cancer types [21]. The product of the *TP53* gene, the p53 protein, is a cell stress-activated sequence-specific transcription factor that regulates gene expression in essential cellular processes [22]. The vast majority of *TP53* mutations occur in exons 5–8 in conserved regions of the DNA binding domain and lead to a formation of a stable mutant protein that loses its tumor suppressive activities, such as the induction of cell cycle arrest, apoptosis, senescence, or DNA repair. In addition, mutant proteins often acquire additional oncogenic functions that provide cells with benefits for growth, invasiveness, and survival [22,23,24].

Regarding colorectal cancer, previous works have shown that mutations in the *TP53* gene occur early in carcinogenesis and are associated with a transition from adenomas with low-grade dysplasia to high-grade dysplasia as well as with the transition from adenoma to carcinoma [25,26,27]. We confirmed this fact in our previous study, where we detected a mutation in the *TP53* gene more than three times more often in early and late carcinomas than in advanced adenomas [28]. It has also been suggested in the past that mutations in different regions of the *TP53* gene may have different effects on p53 protein function and therefore have different prognostic significance in colorectal cancer patients [29,30]. Patients with mutations in exon 5 appear to have a better prognosis than patients with mutations in exons 8 [30] or 7 [29]. However, we did not find any mention that individuals with a mutation in a particular exon were predisposed to a tumor relapse.

For colorectal precancerous lesions, to our knowledge, a study investigating the occurrence of *TP53* gene mutations in index lesions in relation to the incidence of metachronous lesions has not yet been performed and the results of this work are therefore unique. Similar results, but for mutations in the *KRAS* gene, were presented in a study by Juárez et al. [31]. However, we did not confirm the relationship between the presence of a mutation in the *KRAS* gene in the index lesion and the risk of metachronous adenomas in this study.

There is a retrospective study by Speroni et al. that describes the association of the p53 protein expression in colorectal adenomas with the risk of relapse or coexisting adenomas, but this study was performed in a different design to ours [32]. They used an immunohistochemical assay to detect the mutated protein p53, which is based on the fact that while wild-type p53 is present in healthy cells at low concentrations, mutant p53 proteins are often overexpressed and accumulated in the tumor cell nucleus [33,34]. This accumulation is due to an impaired ability to activate the expression of some target genes, e.g., *MDM2*, which mediates the degradation of p53 in proteasomes in the feedback loop [35]. During the Speroni study, the p53 protein accumulation due to *TP53* gene mutation in 100 FFPE samples of endoscopically resected sporadic colon adenomas from 79 patients was immunohistochemically determined and correlated with the occurrence of previous adenomas/carcinomas or coexisting adenomas. The data obtained clearly show that p53 protein expression is a risk factor in this regard, as 52.5% of patients with p53 expressing adenomas (regardless of the degree of dysplasia) had previous adenomas/carcinomas or coexisting adenomas. However, a careful interpretation of the results is warranted, given that overexpression of nuclear p53 can sometimes occur without mutation and vice versa [36,37]. Nevertheless, we want to address the findings of Speroni’s study in future research, which will focus on the detailed determination of mutational heterogeneity of metachronous lesions and its relationship to index and synchronous lesions. We also want to concentrate on explaining the specific location of risk mutations in exon 7 of the *TP53* gene (not in other exons), which has not yet been described in connection with colorectal adenoma/carcinoma and which may be related, for example, to a certain lifestyle of patients.

## 5. Conclusions

Our study utilizing extensive analysis of the mutational burden in colorectal index lesions found a statistically significant association between the mutation in exon 7 of the *TP53* gene and the risk of metachronous adenoma. The findings of our study could lead to an adjustment of existing recommendations for surveillance colonoscopies, possibly in patients with this specific mutation. In this group of patients, shortening the surveillance interval may be considered to decrease the risk of advanced neoplasia emerging after a prolonged follow-up time.

## Figures and Tables

**Figure 1 cancers-14-02823-f001:**
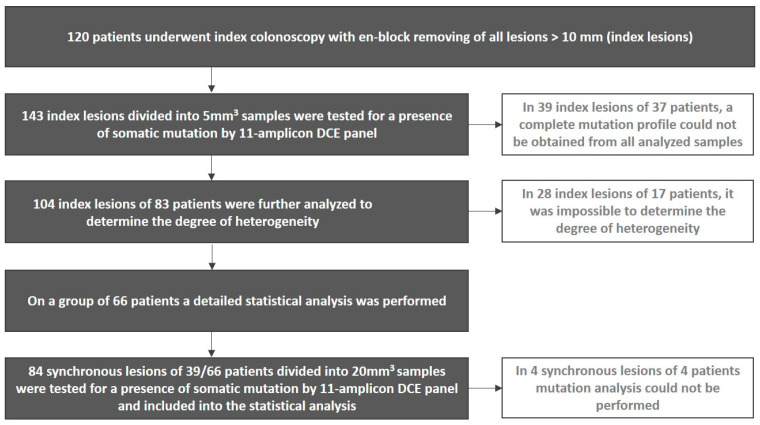
Workflow scheme used for detailed characterization of index and synchronous colorectal lesions.

**Figure 2 cancers-14-02823-f002:**
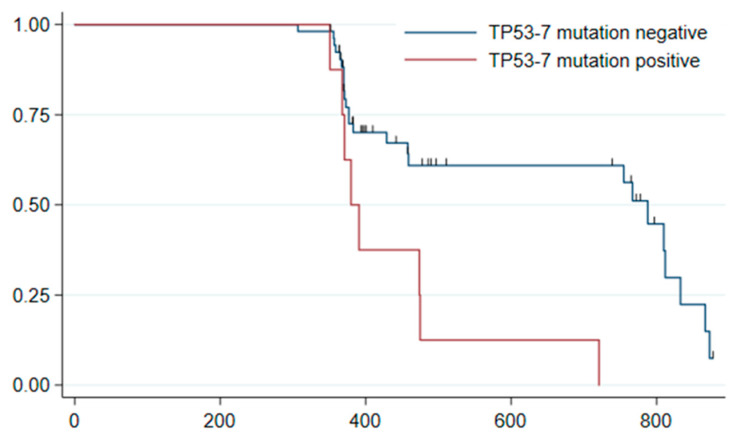
Mutations in exon 7 of the *TP53* gene found in the index lesion as a predictor of metachronous adenoma. Kaplan–Meier survival curves: observed event—detection of adenoma at surveillance colonoscopy in patients with versus without mutation in exon 7 of the *TP53* gene.

**Table 1 cancers-14-02823-t001:** Determination of the degree of heterogeneity in index lesions.

Degree of Heterogeneity	Number of Mutation Clones	Percentage Difference in Mutated Fractions
I	0 or 1	low
II	1	middle
III	1	high
	2	low
IV	2	middle/high
	3	low
V	3	middle/high
	˃3	low/middle/high

**Table 2 cancers-14-02823-t002:** Numbers of mutational clones found in 104 index lesions and 80 synchronous lesions.

Number of Mutation Clones	Number of Index Lesions (%)	Number of Synchronous Lesions (%)
0	27 (26%)	44 (55%)
1	44 (42%)	30 (37.5%)
2	18 (17%)	6 (7.5%)
3	10 (10%)	0 (0%)
4	3 (3%)	0 (0%)
5	2 (2%)	0 (0%)

**Table 3 cancers-14-02823-t003:** Characteristics of patients bearing mutation in exon 7 of the *TP53* gene in index lesion.

Patient ID/Gender	Number of Index Lesions	Mutations Found in Index Lesion(s)	Degree of Heterogeneity	Number of Synchronous Lesions	Mutations Found in Synchronous Lesion(s)	Interval to Surveillance Colonoscopy (Months)	Result of Surveillance Colonoscopy	*TP53* Mutations Found in Metachronous Adenoma(s)
11/M	1x hyperplastic polyp	*BRAF*	1	2	*BRAF*	18	1x sessile serrated lesion, no dysplasia 1x tubular adenoma, LGD	none
	1x hyperplastic polyp	*BRAF*	1					
	1x tubulovillous adenoma, predominantly LGD, in one section HGD	*TP53*-ex7	3					
	1x tubulovillous adenoma, LGD	*APC*	2					
39/M	1x sessile serrated lesion, no dysplasia	*TP53*-ex7, *TP53*-ex8, *TP53*-ex5, *BRAF*, *APC*	5	2	No mutation	12	1x tubular adenoma, LGD, 2x hyperplastic polyp	none
52/M	1x hyperplastic polyp, no dysplasia	*TP53*-ex7, *BRAF*	4	5	*APC*	15	2x tubular adenoma, LGD	none
58/M	1x tubular adenoma, LGD	none	1	2	No mutation	11	3x tubular adenoma, LGD, 1x tubular adenoma, predominantly LGD, sometimes HGD	none
	1x tubular adenoma, LGD	*TP53*-ex7	1					
74/M	1x sessile serrated lesion, no dysplasia	*TP53*-ex7, *TP53*-ex8, *TP53*-ex5, *BRAF*	5	3	*BRAF*, *APC*	12	3x tubular adenoma, LGD	none
86/M	1x tubulovillous adenoma, LGD	*TP53*-ex7, *TP53*-ex8, *TP53*-ex5, *KRAS*	5	3	*APC*	12	1x tubulovillous adenoma, LGD	none
126/F	1x tubulovillous adenoma, LGD	*TP53*-ex7, *TP53*-ex8, *APC*	5	3	*KRAS*, *BRAF*	12	2x tubular adenoma, LGD, 1x hyperplastic polyp	none
134/M	1x tubulovillous adenoma, LGD	*TP53*-ex7	not determined	3	No mutation	15	1x tubular adenoma, LGD	none

M = male; F = female; Ex = exon; LGD = low grade dysplasia; HGD = high grade dysplasia.

## Data Availability

The data presented in this study are available in Appendix A.

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
