# Peer review of "Somatic Mutations in Exon 7 of the TP53 Gene in Index Colorectal Lesions Are Associated with the Early Occurrence of Metachronous Adenoma"

_cancers, 2022, doi:10.3390/cancers14122823_

Round 1

Reviewer 1 Report

The manuscript entitled "Somatic mutations in exon 7 of the TP53 gene in index colorectal lesions are associated with the early occurrence of metachronous adenoma" is primarily focused on showing the importance of somatic mutations in the index lesions to estimate the risk of metachronous lesion recurrence. Overall, the study is well-designed and well-written with supporting analysis. However, the number of samples to derive this very innovative and important conclusion appears to be small.

Author Response

Thank you very much for your positive evaluation. Thank you also for your comment regarding the small number of samples. We are aware of this handicap. As our study brought interesting conclusions and several new questions, we decided to expand it and continue our research. An extension study is already in process, which should, among other things, confirm the initial results on a larger number of samples.

Reviewer 2 Report

  • Identifying patients with an increased risk of early recurrence of colorectal lesions through the genotype about colorectal neoplasia during index colonoscopy may helpful the further therapy.
  • Except the 8 patients with mutation in exon 7 of the TP53 gene, are this mutation found in synchronous lesions of other patients? Or in meta-chronousad-enomas beside these 8 patients?
  • It is better to show the mutation of all 11 genes loci in all patients?

Author Response

Dear referee,
thank you very much for your evaluation and comments. Below are answers to your questions:

  • Except the 8 patients with mutation in exon 7 of the TP53 gene, are this mutation found in synchronous lesions of other patients? Or in meta-chronous adenomas beside these 8 patients?

Thank you for this point. Interestingly, there was no mutation in exon 7 of the TP53 gene in any synchronous lesion, as stated in the last sentence of the result section. For clarification the following has been added to the last sentence: "of any of the 66 patients“.

We tested only presence of mutations in the TP53 gene in metachronous adenomas of the eight patients with a mutation in exon 7 of the TP53 gene found  in the index lesions. The main purpose was to confirm that the tested metachronous adenomas were not in fact fast-growing lesions overlooked by index colonoscopy (interval lesions). However, this analysis was already performed beyond the original study. An extension study is already in process, which focus, among other things, on detailed mutation analysis of the metachronous adenomas.

  • It is better to show the mutation of all 11 genes loci in all patients?

We agree that it would be a good idea to show a table of all the mutations, but this would be quite extensive and reader would probably find it difficult to understand. Here we enclose a simplified version containing a final set of 66 patients and we will leave it up to you to consider whether to include it in the article.

Regarding the extensive English revision - our manuscript was read by several English-speaking colleagues and we also had the text reviewed by a native expert speaker before submission. We have sent a proof of the language correction to the editorial office. Additionally, minor grammar changes in the current version of manuscript have been made.
